# Human umbilical cord-derived mesenchymal stem cells alleviate autoimmune hepatitis by inhibiting hepatic ferroptosis

Yaqin Li[1,2], Bing Liu[3], Guoxin Hu[2], Tao Zhou[4], Xuanqiu He[2], Ling Guo[2], Luoshi Zhang[1], Weizhao Tong[2], Yihua Chen[2], Youhua Xu[1]*, Guangdong Tong[1,5]*, Wei V. Zheng[4]*

1 Faculty of Chinese Medicine, Macau University of Science and Technology, Taipa, Macao, People's Republic of China, 2 Department of Infectious Disease, Peking University Shenzhen Hospital, Shenzhen, Guangdong, People's Republic of China, 3 Department of Hematology, Peking University Shenzhen Hospital, Shenzhen, Guangdong, People's Republic of China, 4 Intervention and Cell Therapy Center, Peking University Shenzhen Hospital, Shenzhen, Guangdong, People's Republic of China, 5 Department of Liver Disease, Shenzhen Traditional Chinese Medicine Hospital, The Fourth Clinical Medical College of Guangzhou University of Chinese Medicine, Shenzhen, Guangdong, People's Republic of China

* zhengw2013@yeah.net (WVZ); tgd755@163.com (GT); yhxu@must.edu.mo (YX)

## Abstract

### Background

Autoimmune hepatitis (AIH) is a liver disease marked by immune-mediated hepatocyte damage. Current treatments have variable patient responses and considerable side effects, highlighting the need for alternative therapies. Human umbilical cord-derived mesenchymal stem cells (hUC-MSCs) have shown therapeutic potential in liver diseases, but their mechanisms in AIH remain unclear.

### Methods

A Concanavalin A (ConA)-induced AIH-like mouse model was used to assess the therapeutic effects of hUC-MSCs. Survival, liver function-related serum marker expression, histopathology, and apoptosis were evaluated. Metabolomic profiling and ferroptosis-related markers were analyzed to uncover potential mechanisms.

### Results

In ConA-induced AIH-like mouse model challenged with a lethal dose of ConA, hUC-MSC treatment significantly ameliorated liver tissue damage and serum liver function parameters, alleviated hepatocyte apoptosis, and improved survival. Metabolomic and ontology analyses of mouse liver tissue samples revealed that hUC-MSCs treatment altered the levels of metabolites (Glu derivatives and peptides) functionally associated with ferroptosis-related pathways. hUC-MSCs partially reversed ConA-induced malondialdehyde (MDA), oxidized glutathione (GSSG), glutamate, and $Fe^{2+}$, while restoring reduced glutathione (GSH). Expression of COX2 was downregulated,

**Data availability statement:** All relevant data are within the manuscript and its Supporting information files.

**Funding:** This work was supported by National Natural Science Foundation of China [82371568, 82570728], Shenzhen Science and Technology Program [JCYJ20220818102818039], Shenzhen Biomedical Industry Major Public Service Platform and Core Technology Research Special Support Plan [XMHT20220104048], BYD Charity Foundation [PUSH-BYD JCYJ20220401, PUSH-BYD JCYJ20220], Medical Scientific Research Foundation of Guangdong Province [A2024220] and Peking University Shenzhen Hospital Research Foundation [JCYJ 2021014].

**Competing interests:** No authors have competing interests.

whereas key ferroptosis suppressors, SLC7A11, GPX4, and FTH1, were upregulated following hUC-MSC treatment.

## Conclusions

Based on the above evidence, we propose that hUC-MSCs may ameliorate ConA-induced liver injury in mice, potentially through modulation of ferroptosis-related pathways, and we support further investigation of hUC-MSCs as potential treatments for AIH. We believe that further in-depth studies are still needed to elaborate on the detailed regulatory mechanisms of MSCs on the ferroptosis pathway during the treatment of AIH.

## 1. Introduction

Autoimmune hepatitis (AIH) is a chronic condition in which the immune system mistakenly targets and damages liver tissue [1]. Although the exact cause remains unknown, factors such as genetic susceptibility and infections have been linked to its development [2]. AIH can progress to serious liver complications, including cirrhosis, liver failure, and hepatocellular carcinoma [2]. Globally, the annual incidence and prevalence are estimated at 1.37 and 17.44 cases per 100,000 people, respectively, with notably higher rates in Europe and North America [3].

The current treatment approach for AIH typically involves corticosteroids to induce remission, followed by long-term use of nonsteroidal immunosuppressive drugs for maintenance [4,5]. However, treatment effectiveness is often limited due to variable patient responses and considerable side effects. Moreover, little therapeutic advancement has been made in recent years [6], underscoring the urgent need for novel and more effective therapies.

Among emerging strategies, stem cell-based therapies have gained attention for their potential to treat various diseases [7,8]. In particular, mesenchymal stem cells (MSCs), including those derived from hUC-MSCs, have shown promising regenerative and immunomodulatory properties [9,10]. Early studies suggest that MSCs may be beneficial in treating autoimmune liver disorders, including AIH [11]. However, the specific mechanisms by which MSCs exert protective effects on the liver remain incompletely understood.

One potential mechanism involves ferroptosis, a distinct form of regulated cell death characterized by iron accumulation, lipid peroxidation, and oxidative stress. Unlike apoptosis or necrosis, ferroptosis results from excessive reactive oxygen species (ROS) and reduced glutathione levels, culminating in cellular membrane damage and death [12,13]. Recent findings indicate a role for ferroptosis in AIH pathogenesis. For instance, mouse models of AIH induced with Concanavalin A (ConA) show increased ferroptotic activity, which can be mitigated by fibroblast growth factor 4 through activation of CISD3 [14]. These insights suggest that targeting ferroptosis might offer a new therapeutic direction for AIH. However, it remains uncertain whether MSCs can confer liver protection by modulating ferroptosis.

To investigate this, we established a ConA-induced AIH-like mouse model and treated the animals with hUC-MSCs. The results demonstrated that hUC-MSCs significantly improved survival rates and liver function in AIH-like mice. Subsequent metabolomic analysis pointed to ferroptosis as a key pathway influenced by MSC treatment. To explore this further, we assessed multiple ferroptosis-associated markers to determine whether MSCs mitigate liver damage by inhibiting this form of cell death.

## 2. Materials and methods

### 2.1. MSC isolation and characterization

The umbilical cord was disinfected in 75% ethanol (1 minute), thoroughly washed with saline, and trimmed at both ends. After repeated washing until the effluent became clear, it was sectioned into 2 cm segments, and Wharton's jelly was isolated via careful dissection. The harvested tissue was minced, centrifuged ($300 \times g$, 8 min), and evenly plated in eight 150 cm$^2$ dishes containing MSC serum-free culture medium (#NC0107, Yocon, China), followed by incubation at 37°C under 5% $CO_2$ in a humidified atmosphere. This study was conducted in accordance with the ethical principles of the Declaration of Helsinki. The experimental protocols involving human participants were reviewed and approved by the Ethics Committee of Peking University Shenzhen Hospital (Approval No. 017 [2023]). Prior to the study, written informed consent was obtained from all individual participants.

To assess the viability and identity of MSCs, an aliquot of the MSC suspension was stained with trypan blue and analyzed using a Countess 3™ automated cell counter (Invitrogen, USA). For immunophenotyping, another aliquot was stained with individual primary antibodies and analyzed by flow cytometry on a NovoCyte 1300 instrument (Agilent, USA). Briefly, cells ($1 \times 10^5$ per tube) were washed with PBS, incubated with the indicated antibodies for 30 min at 4 °C in the dark, washed twice, and analyzed. Isotype controls were used to define positivity thresholds. The characterization followed the ISCT criteria [15]. The following FITC-conjugated mouse anti-human antibodies (BioLegend, USA) were used: CD73 (AD2, #344016), CD90 (5E10, #328108), CD105 (43A3, #323204), CD11b (ICRF44, #301404), CD19 (HIB19, #302206), CD34 (581, #343504), CD45 (HI30, #304006), and HLA-DR (L243, #307604).

For MSC labeling, lentivirus carrying the pLVX-IRES-ZsGreen-Luciferase construct was generated using the Lenti-X™ HTX Packaging System (#631247, Takara, Japan) and the Lenti-X 293T Cell Line (#632180, Takara) according to the manufacturer's instructions. MSCs were seeded in 6-well plates at a density of $2 \times 10^6$ cells/well and cultured for 6 hours. After rinsing with PBS, the cells were exposed to lentivirus for 24 hours, followed by culture in BC-T4 medium supplemented with 10% fetal bovine serum (FBS, #10082147, Gibco, USA) for an additional two days before further analysis.

### 2.2. Animal model establishment and treatment

Concanavalin A (ConA, Cat. #L7647) was obtained from Sigma-Aldrich (USA). C57BL/6 female mice (6−8 weeks old, approximately 22−25 grams) were obtained from the Animal Center of Guangdong PharmaKlean Biological Technology Co., Ltd. (China). The mice were then randomly assigned into four groups (n = 6 per group): untreated control (NC), MSC-only (MSC), ConA-only+PBS (ConA), and ConA combined with MSC treatment (ConA+MSC). AIH-like phenotypes were induced by intravenous injection of ConA at 20 mg/kg. hUC-MSCs were administered via the tail vein at $1 \times 10^6$ cells per mouse. For combination treatment, hUC-MSCs were injected 1.5 hours after ConA administration. Twenty-four hours post-ConA administration, blood samples were collected from the orbital venous plexus of mice in each group, followed by euthanasia via cervical dislocation. The blood samples were centrifuged at 3000 rpm for 10 minutes, and the resulting supernatants were stored at −80°C. Simultaneously, liver tissues were dissected and photographed. Half of each liver was fixed in 4% paraformaldehyde for histological analysis, while the remaining tissues were stored for Western blotting and biochemical assays. All animal experiments were conducted in accordance with the guidelines approved by the Ethics Committee of Peking University Shenzhen-Hong Kong University of Science and Technology Medical Center (Approval No. 2022-766; August 2022).

## 2.3. Hematoxylin and Eosin (H&E) staining and TUNEL

Paraffin-embedded liver sections were deparaffinized and rehydrated using xylene and graded ethanol. H&E staining was performed using a commercial kit (C0105S, Beyotime, China). Apoptosis was assessed using a TUNEL assay kit (KGA1406−100, KeyGEN BioTECH, China), according to the manufacturer's instructions. Images were captured using a BX51 microscope (Olympus, Japan).

## 2.4. Metabolomic analysis

Liver tissues from three mice per group were subjected to metabolomic profiling by GENE DENOVO Biotechnology (China). Approximately 100 mg of tissue was ground in liquid nitrogen, resuspended in pre-chilled 80% methanol with 0.1% formic acid, vortexed, and incubated on ice for 5 minutes. Samples were centrifuged at 15,000 g at 4 °C for 20 minutes. The supernatant was diluted to 53% methanol with LC-MS grade water, centrifuged again, and the final supernatant was injected into the LC-MS/MS system.

Liquid chromatography–mass spectrometry (LC-MS/MS) was conducted using an ExionLC™ AD system coupled to a QTRAP® 6500 + mass spectrometer (SCIEX, Canada). Chromatographic separation was performed on an Xselect HSS T3 column (2.1 × 150 mm, 2.5 μm) with a 20-minute linear gradient at 0.4 mL/min in both positive and negative ion modes. The mobile phases were 0.1% formic acid in water (eluent A) and 0.1% formic acid in acetonitrile (eluent B), with the gradient profile as follows: 2% B for 2 min, ramp to 100% B over 15 min, hold for 2 min, return to 2% B in 0.1 min, and re-equilibrate for 0.9 min.

Instrument parameters were set as follows: curtain gas 35 psi, collision gas medium, temperature 550 °C, ion spray voltage ±5500 V (positive/negative modes), and ion source gases 1 and 2 both at 60 psi. Metabolite identification and quantification were conducted using the Multiple Reaction Monitoring (MRM) method. SCIEX OS software (v1.4) was used for peak integration and correction. Quantification was based on Q3 peak area, and identification employed Q1, Q3, retention time, declustering potential, and collision energy. The minimum peak height was set to 500, the signal-to-noise ratio to 5, and the Gaussian smoothing width to 1.

## 2.5. Orthogonal projection to latent structures-discriminant analysis (OPLS-DA)

Discriminant analysis was performed using OPLS-DA [16] in R (v4.3.0). Model validation was conducted through seven-fold cross-validation and 200-permutation tests [17]. The R2 value indicated the proportion of variation explained by the model, while the Q2 value evaluated the model's predictive ability. In general, Q2 values > 0.4 were considered indicative of acceptable predictive quality, and Q2 values > 0.9 denoted excellent model robustness.

## 2.6. Identification of differentially expressed metabolites (DEMs)

To identify DEMs, both multivariate and univariate statistical approaches were applied. Variable Importance in Projection (VIP) scores derived from the OPLS-DA model were used to rank the contribution of each metabolite to group separation, with a threshold of VIP > 1. In parallel, unpaired t-tests were conducted, and metabolites with p-values < 0.05 were considered statistically significant. Only metabolites meeting both criteria (VIP > 1 and p-value < 0.05) were classified as significant DEMs. These DEMs were normalized using z-score transformation and visualized via hierarchical clustering with the R package 'ComplexHeatmap' (v2.16.0). The Venn diagram was generated by Venny2.1 (https://bioinfogp.cnb.csic.es/tools/venny/).

## 2.7. qPCR

Total RNA was extracted from liver tissues using the RaPure Total RNA Kit (R4011-02, Magen, China) following the manufacturer's protocol. Complementary DNA (cDNA) was synthesized using the All-in-One First-Strand SuperMix (NY-Bio,

China). Quantitative real-time PCR (qPCR) was performed using the ChamQ SYBR qPCR Master Mix (21–040-CV, Vazyme, China) on the QuantStudio 3 Real-Time PCR System (A28573, Thermo Fisher Scientific, USA). Gene expression levels were calculated using the $2^{-\Delta\Delta Ct}$ method, with 18S rRNA used as the internal control for normalization. The following qPCR primers were used: 18S-qF: 5'- CGACGACCCATTCGAACGTCT-3',18S-qR: 5'-CTCTCCGGAAT CGAA CCCTGA-3'; COX2-qF: 5'-CTGGCGCTCAGCCATACAG-3', COX2-qR: 5'- CGCACTTATACTGGTCAAATCCC-3'; SLC7A11-qF: 5'- TCTCCAAAGGAGGTTACCTGC-3', SLC7A11-qR: 5'- AGACTCCCCTCAGTAAAGTGAC-3'; GPX4-qF: 5'- ACAAGAACGGCTGCGTGGTGAA-3', GPX4-qR: 5'- GCCACACACTTGTGGAGCTAGA-3'; FTH1-qF: 5'- GAACTA CCACCAGGACTCAGA-3', FTH1-qR: 5'- TAAACGTAGGAGGCGTAGAGC-3'; Luciferase-qF: 5'- AGCTTCTTCGCT AAGAGCACT-3', Luciferase-qR: 5'- GCTGGTTGTTTCTGTCAGGC-3'; ZsGreen-qF: 5'-TTCATGTACGGCAACCGC-3', ZsGreen-qR: 5'-TCGGCGTTGCAGATGCACA-3'.

## 2.8. Western blot

Liver proteins were extracted using RIPA lysis buffer supplemented with protease inhibitors (Beyotime), following the manufacturer's protocol. Lysates were centrifuged at 13,000 × g for 15 minutes at 4 °C, and the supernatants were collected and denatured. Protein concentrations were determined using the Pierce BCA Protein Assay Kit (Thermo Fisher Scientific). Equal amounts of protein were separated by 10% SDS-PAGE and transferred onto PVDF membranes. Membranes were blocked with 5% non-fat milk for 1 hour at room temperature (RT) and then incubated overnight at 4 °C with primary antibodies. After washing with TBST, membranes were incubated with HRP-conjugated secondary antibodies for 2 hours at RT. Protein bands were detected using ECL reagents and visualized with X-ray film. Band intensities were quantified using ImageJ software (NIH, USA). Antibodies used are detailed in Table 1.

## 2.9. Biochemical measurement of liver function

Serum levels of alkaline phosphatase (ALP), alanine aminotransferase (ALT), aspartate aminotransferase (AST), total bilirubin (TBIL), direct bilirubin (DBIL), indirect bilirubin (IBIL), and total bile Acids (TBA) were measured using commercial kits from Melkang (China), including Alkaline Phosphatase Kit (20142400118), Alanine Aminotransferase Kit (20142400109), Aspartate Aminotransferase Kit (20142400119), Total Bilirubin Kit (20142400132), Direct Bilirubin Kit (20142400121), and Bile Acids Kit (20142400115). All assays were performed according to the manufacturers' instructions using a HITACHI-3110 analyzer.

## 2.10. Assessment of lipid peroxidation, glutathione, Glu, and $Fe^{2+}$ levels

Malondialdehyde (MDA), an indicator of lipid peroxidation, was measured using a kit from Solarbio (China). Cell lysates were incubated with the provided reagent and heated at 95°C for 40 minutes. The absorbance at 532 nm ($OD_{532}$) was recorded, and MDA concentrations were calculated using a standard curve.

**Table 1. Western blot antibodies.**

| Target | Manufacturer | Catalog number | Dilution |
|---|---|---|---|
| COX2 | Proteintech (China) | 27308-1-AP | 1:500 |
| SLC7A11 | Proteintech | 26864-1-AP | 1:500 |
| FTH1 | Proteintech | 83428-1-RR | 1:500 |
| GPX4 | Proteintech | 67763-1-Ig | 1:500 |
| Actin | Proteintech | 20536-1-AP | 1:1000 |
| HRP-Anti-Rabbit IgG | Proteintech | SA00001-2 | 1:10000 |

Reduced (GSH) and oxidized glutathione (GSSG) levels were determined using a GSH/GSSG assay kit (S0053, Beyotime). Liver lysates were processed following the manufacturer's protocol. $OD_{412}$ was recorded for both GSH and GSSG measurements, and concentrations were calculated from standard curves. The GSH/GSSG ratio was also determined. Glu levels were determined using a kit (ADS-W-AJS007, MEIMIAN, China) following the manufacturer's guidelines.

$Fe^{2+}$ levels were measured using an iron colorimetric assay kit (E-BC-K881-M, Elabscience, USA). Liver tissues were lysed and centrifuged, and the supernatants were incubated with an iron probe to form a colored complex. $OD_{593}$ values were measured after incubation at 37°C for 10 minutes. $Fe^{2+}$ concentrations were calculated from a standard curve and normalized to total protein levels (determined by BCA assay).

### 2.11. Statistical analysis

All data were analyzed using GraphPad Prism 8, except where bioinformatic tools were applied. Results are presented as mean ± standard error of the mean (SEM). Statistical significance was assessed using the methods indicated in the figure legends. A p-value < 0.05 was considered statistically significant.

## 3. Results

### 3.1. hUC-MSCs enhance survival and liver function in AIH-like mice

To determine the therapeutic potential of hUC-MSCs in AIH, MSCs were isolated from human umbilical cord tissues, and their viability was quantified using an automated cell counter. The analysis showed that 99% of the cells were viable (S1A Fig). Immunophenotyping by flow cytometry further confirmed their MSC identity, revealing uniform expression of canonical MSC markers CD73 (99.6%), CD90 (99.9%), and CD105 (97.4%), while hematopoietic lineage markers CD11b (0.04%), CD34 (0.24%), CD45 (0.95%), HLA-DR (1.09%), and CD19 (1.18%) were largely absent (S1B Fig). We then evaluated their impact on the survival and liver function of mice subjected to ConA-induced liver injury using both lethal (20 µg/g) and working doses (8 µg/g), with ConA (1 mg/mL) administered via tail vein injection. Survival analysis revealed that all mice in the NC, MSC, and ConA+MSC groups remained alive throughout the 24-hour observation period. In contrast, all mice in the ConA-only group succumbed within 18 hours post-injection (Fig 1A).

Gross morphological examination showed that ConA administration caused visible liver damage, including pronounced swelling and redness, which were notably ameliorated following hUC-MSC treatment (Fig 1B). Serum biochemical assays further confirmed liver injury mitigation by hUC-MSCs. Levels of liver function markers, including ALP, ALT, AST, TBIL, DBIL, IBIL, and TBA, were significantly elevated in ConA-treated mice and were substantially reduced upon hUC-MSC administration (Fig 1C–I).

Histological examination using H&E staining revealed widespread tissue damage in ConA-treated livers, characterized by hepatocyte death, which was attenuated in mice receiving hUC-MSCs (Fig 2A). This protective effect was corroborated by TUNEL staining, which demonstrated a marked decrease in hepatocyte apoptosis in the hUC-MSC-treated group compared to ConA alone (Fig 2B).

To confirm the hepatic homing capacity of hUC-MSCs, we transduced the cells with a lentivirus encoding ZsGreen and Luciferase (S2A,B Fig). qPCR analysis demonstrated detectable expression of ZsGreen and Luciferase mRNA in mice injected with transduced hUC-MSCs, with significantly higher levels observed in ConA-treated liver tissues (S2C,D Fig), indicating enhanced recruitment of MSCs to injured livers. Collectively, these results demonstrate that hUC-MSCs significantly enhance survival and improve liver function in the ConA-induced AIH-like mouse model.

### 3.2. Metabolomic profiling reveals pathways influenced by hUC-MSC treatment in ConA-injured liver tissue

To explore the underlying mechanisms of hUC-MSC-mediated hepatoprotection, we performed unbiased metabolomic analysis, leveraging the liver's central role in metabolism. OPLS-DA revealed distinct metabolomic signatures among the

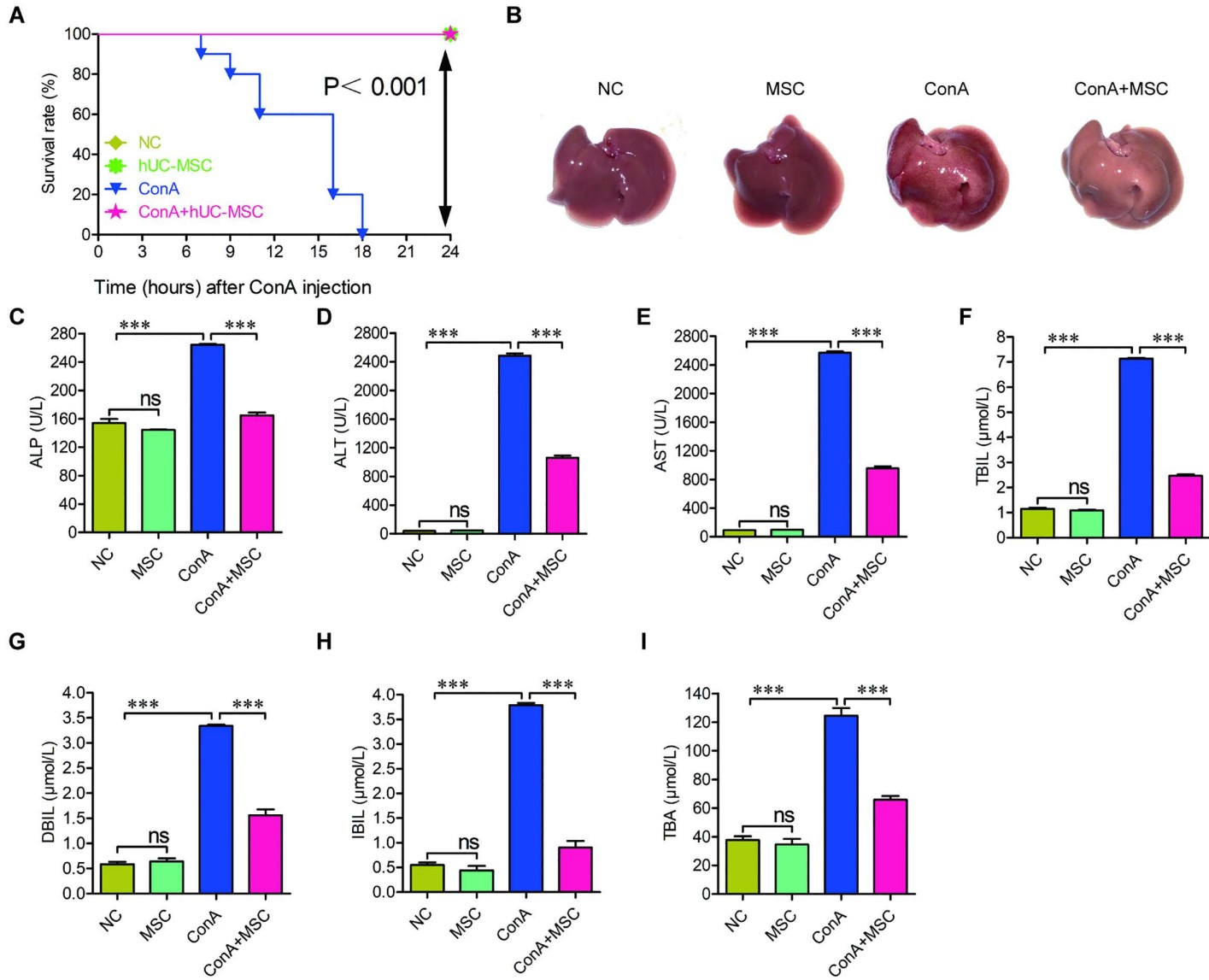

**Fig 1. hUC-MSCs improve survival and liver function in ConA-induced AIH-like mice.** (A) Survival curves of mice in each treatment group within 24 hours post-ConA administration. (B) Gross morphology of livers from each group, showing ConA-induced swelling and discoloration alleviated by hUC-MSCs. (C-I) Serum biochemical markers (ALP, ALT, AST, TBIL, DBIL, IBIL, TBA) reflecting liver function across groups. n=6; *p < 0.05, **p < 0.01, ***p < 0.001. Two-way ANOVA followed by Tukey's HSD was used to calculate the p-values.

NC, MSC, ConA, and ConA+MSC groups (Fig 3A). Comparative analysis identified 263 DEMs between MSC and NC (120 downregulated, 143 upregulated), 454 DEMs between ConA and NC (228 downregulated, 226 upregulated), and 140 DEMs between ConA and ConA+MSC (85 downregulated, 55 upregulated) (Fig 3B). Pathway enrichment analysis using the KEGG highlighted distinct biological pathways associated with these DEMs. In the MSC vs. NC comparison, key pathways included pyruvate metabolism and oxidative phosphorylation (Fig 4A). In the ConA vs. NC group, significant enrichment was observed in pathways such as ABC transporters, nucleotide metabolism, and ferroptosis (Fig 4B). In the ConA vs. ConA+MSC group, enriched pathways included ABC transporters, nucleotide metabolism, vitamin digestion and absorption, and others (Fig 4C).

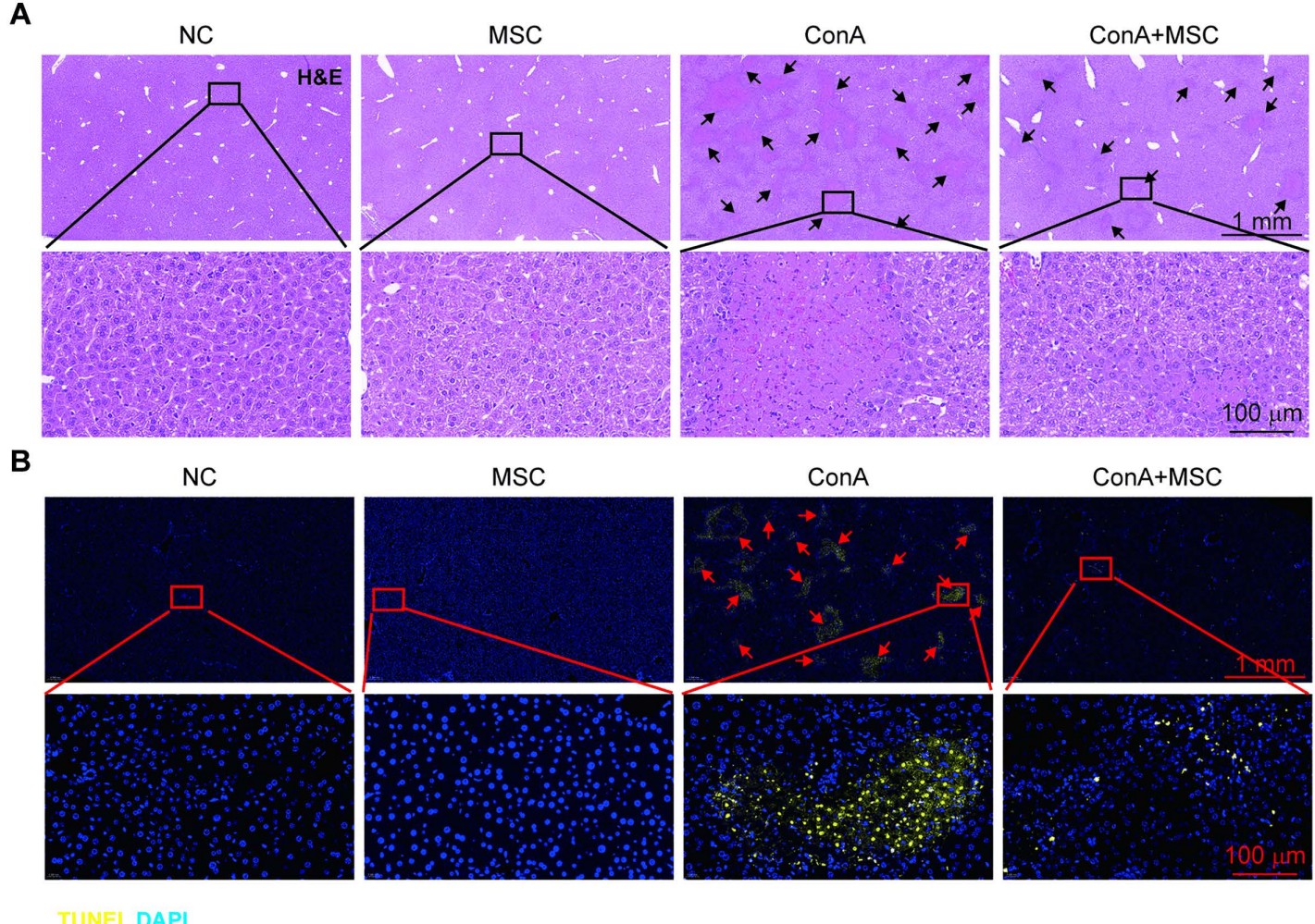

TUNEL DAPI

**Fig 2. hUC-MSCs alleviate hepatic damage and apoptosis in ConA-injured livers.** (A) H&E staining of liver sections showing histopathological damage (black arrows) in ConA-injured livers, which is reduced by hUC-MSC treatment. (B) TUNEL staining indicating apoptotic cells in ConA-damaged liver tissues (red arrows); hUC-MSCs attenuate apoptosis.

Intersection analysis across the three groups revealed two commonly affected pathways (Fig 5A), with associated metabolites classified under Glu derivatives and peptides, as indicated by ontology analysis (Fig 5B). Importantly, these shared DEMs are functionally connected to ferroptosis (Fig 5C), suggesting that modulation of ferroptotic processes may play a role in the therapeutic effects of hUC-MSCs.

### 3.3. hUC-MSC treatment suppresses ferroptosis in ConA-induced liver tissue

To validate whether hUC-MSCs modulate ferroptosis in AIH, we measured several biochemical markers associated with this cell death pathway. ConA-treated mice showed significant elevations in MDA, GSSG, Glu, and $Fe^{2+}$ levels, along with decreased levels of GSH, a profile consistent with enhanced ferroptotic activity. Notably, these alterations were partially reversed in mice receiving hUC-MSC therapy (Fig 6A–F).

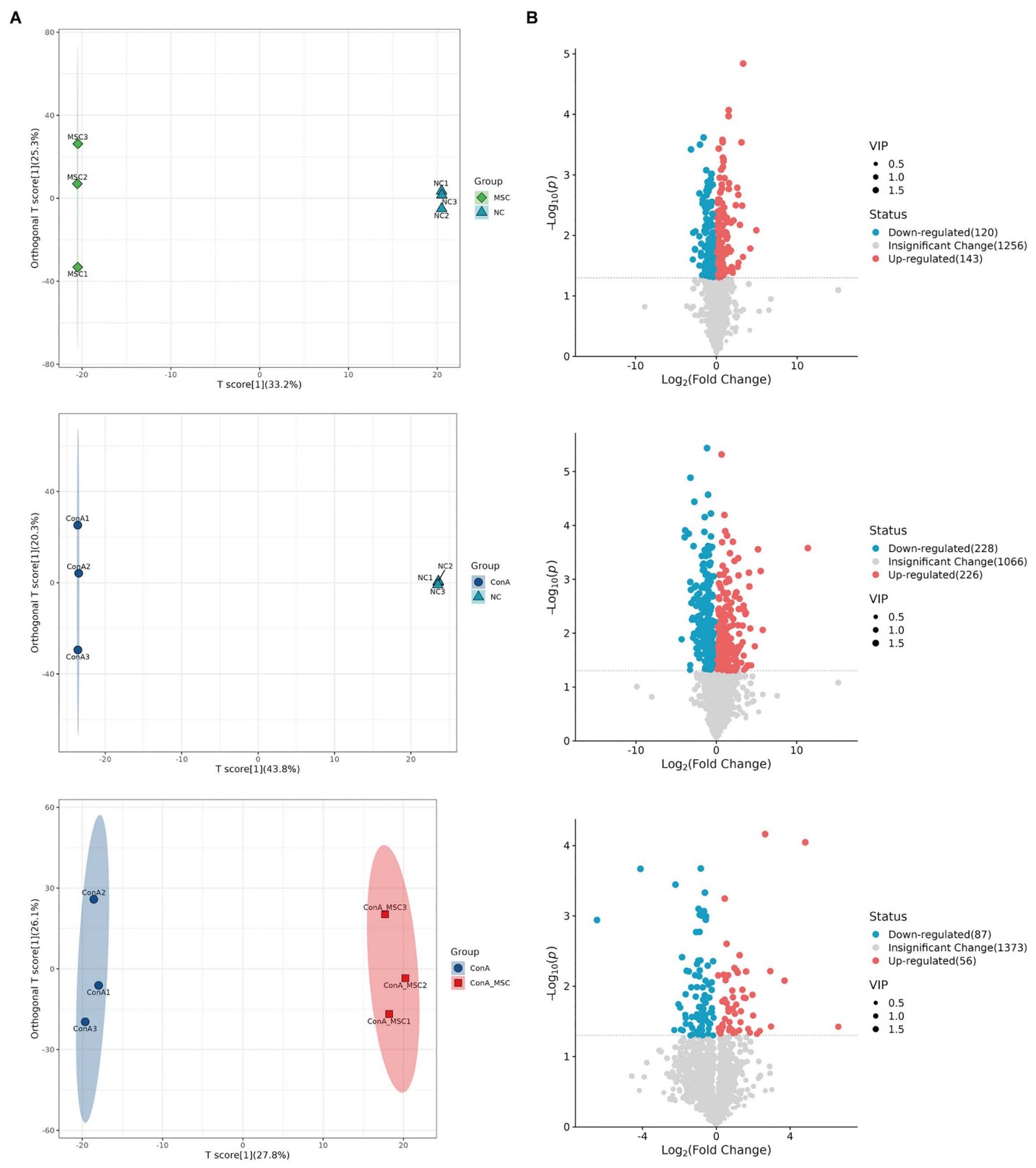

**Fig 3. OPLS-DA reveals distinct metabolomic profiles in liver tissues across treatment groups.** (A) OPLS-DA score plots showing metabolic distinctions among NC, MSC, ConA, and ConA+MSC groups. Each circle or triangle represents one sample (n=3 per group). (B) Volcano plots showing DEMs among the groups. Blue, red, and grey dots represent downregulated, upregulated, and not significantly changed metabolites, respectively.

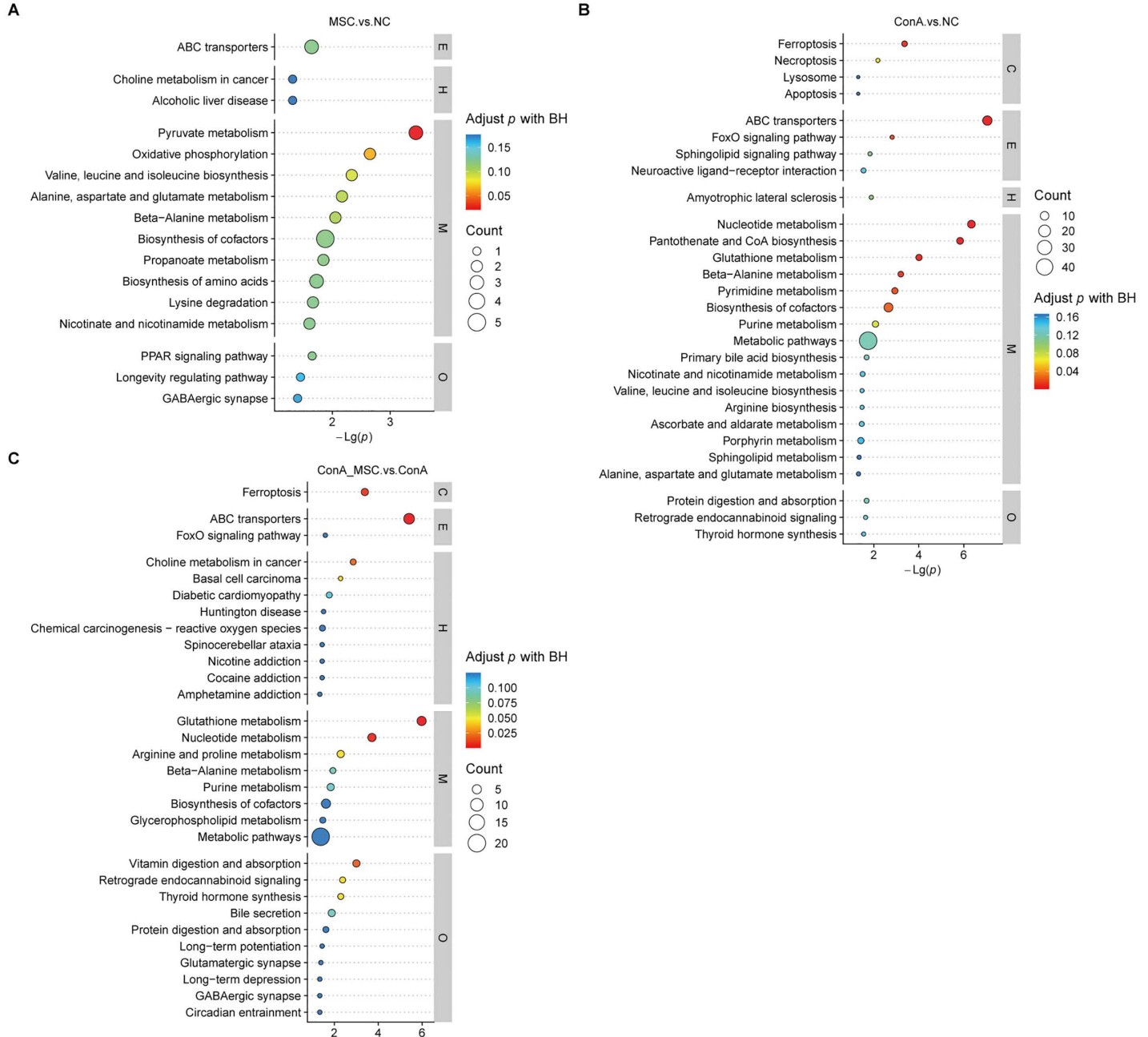

**Fig 4. KEGG pathway enrichment of DEMs between groups.** (A-C) Bubble plots showing the significantly enriched KEGG pathways for DEMs between MSC vs NC (A), ConA vs NC (B), and ConA+MSC vs ConA (C).

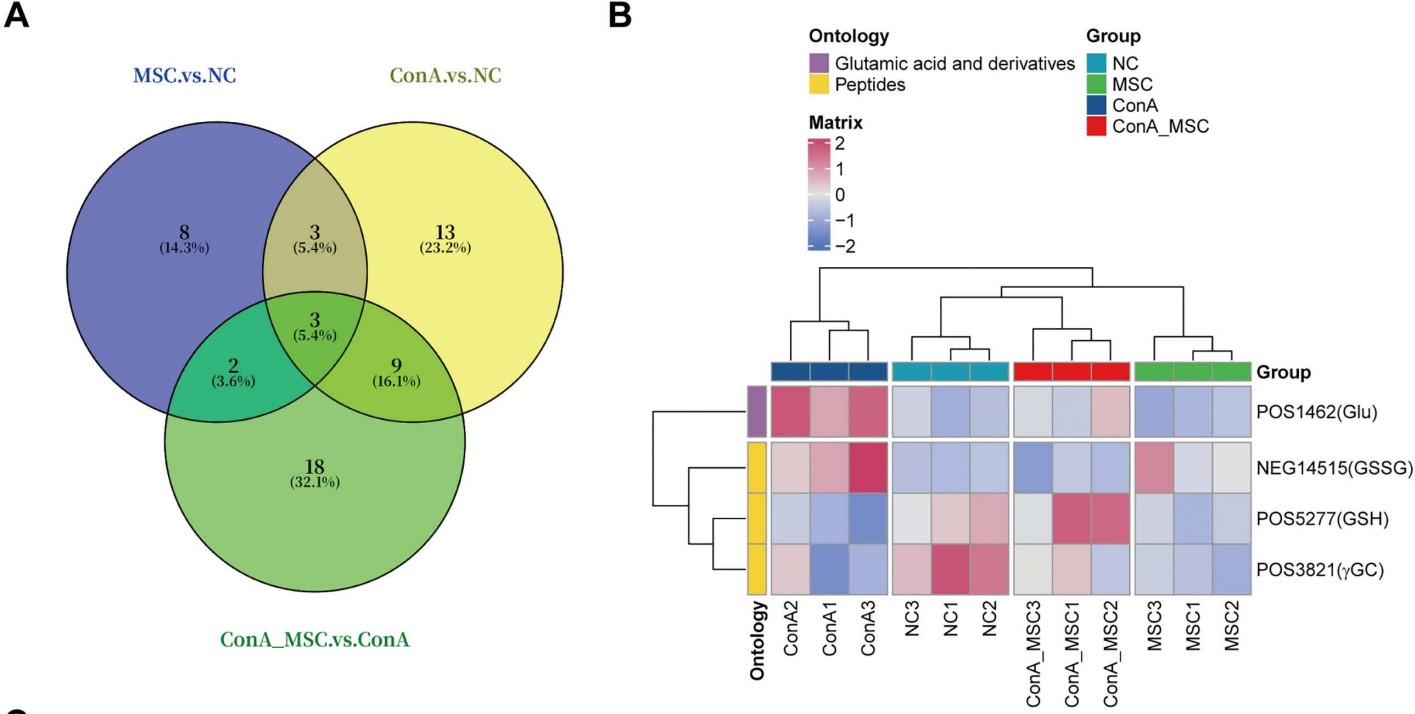

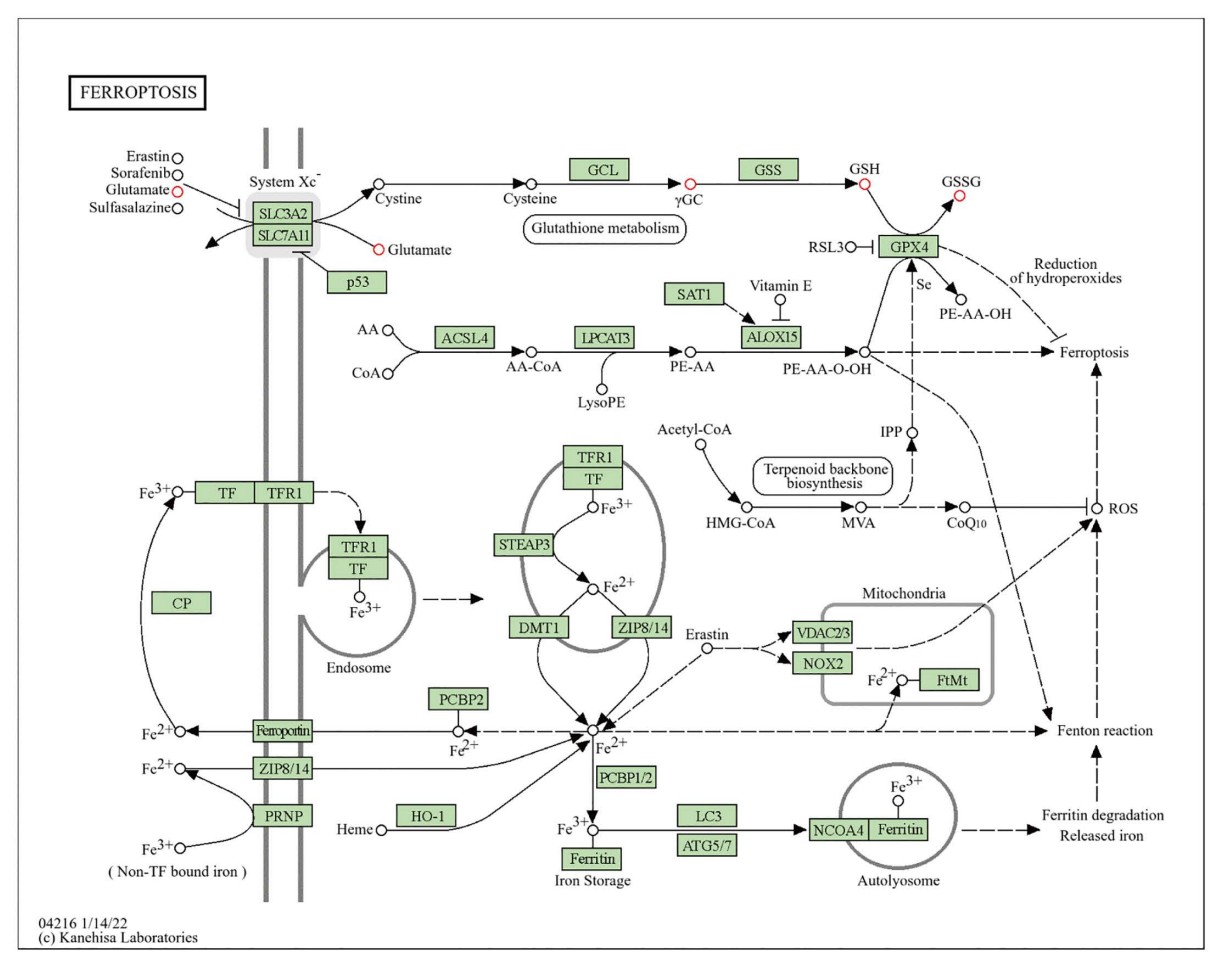

**Fig 5. Overlapping ferroptosis-related metabolic changes regulated by hUC-MSCs.** (A) Venn diagram showing overlapping KEGG pathways among different comparison groups. (B) Ontology classification of DEMs associated with the overlapping pathways. (C) The KEGG map showing the position of four glutamate metabolism components (red circles) in the ferroptosis pathway.

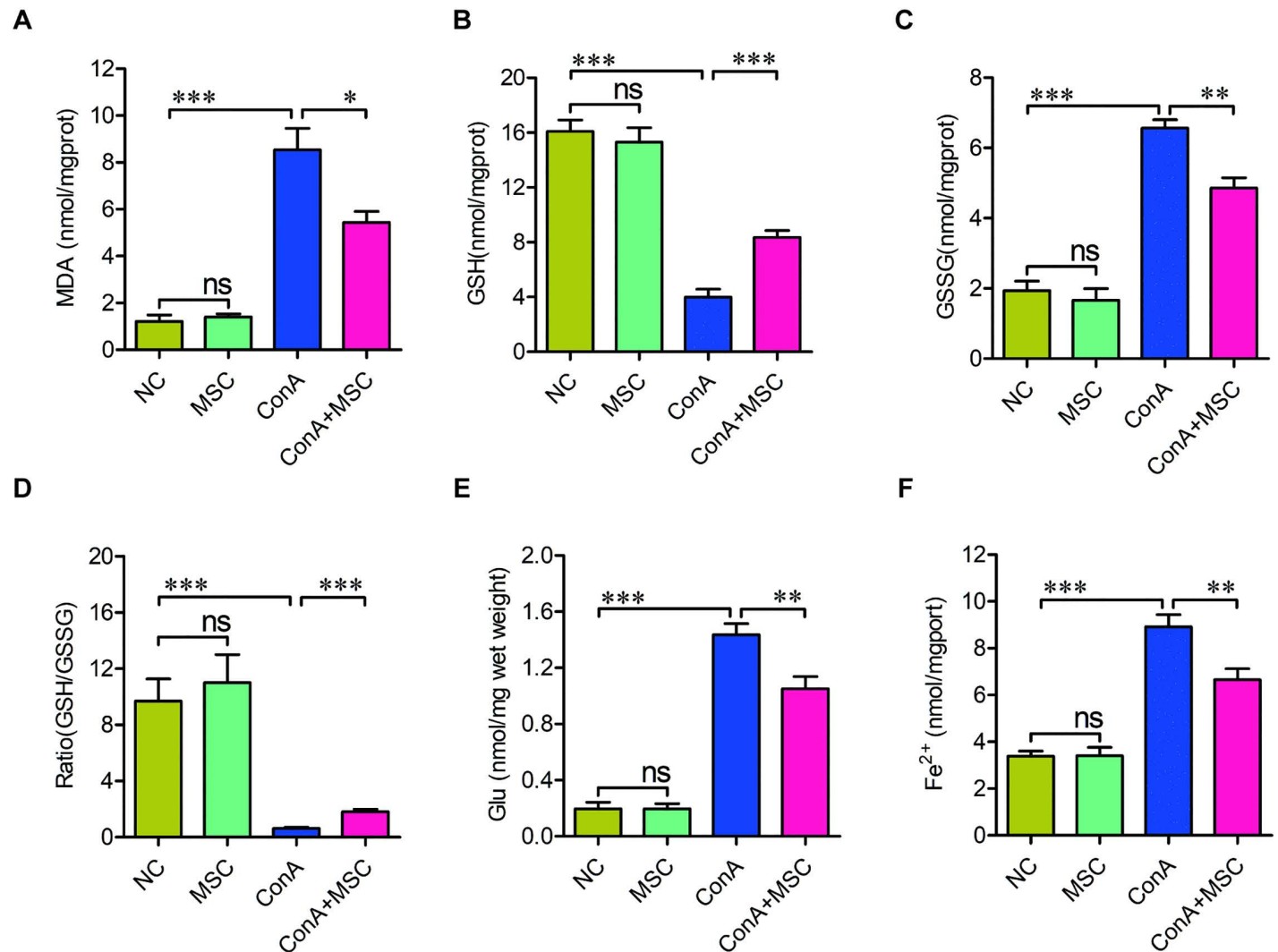

**Fig 6. hUC-MSCs reverse ferroptosis-related biochemical changes in ConA-injured livers.** (A-F) Quantification of MDA (A), GSH (B), GSSG (C), GSH/GSSG ratio (D), Glu(E), and $Fe^{2+}$ (F) levels in liver tissues. n = 6; *p < 0.05, **p < 0.01, ***p < 0.001, ns, not significant. Two-way ANOVA followed by Tukey's HSD was employed to determine the statistical significance.

At the molecular level, qPCR and Western blot analyses revealed increased expression of the ferroptosis marker COX2 and decreased levels of ferroptosis suppressors, including SLC7A11, GPX4, and FTH1, in the ConA group. hUC-MSC treatment effectively mitigated these changes, partially restoring the expression of key anti-ferroptotic proteins (Fig 7A–C). These findings strongly indicate that hUC-MSCs protect against ConA-induced liver injury at least in part by suppressing ferroptosis.

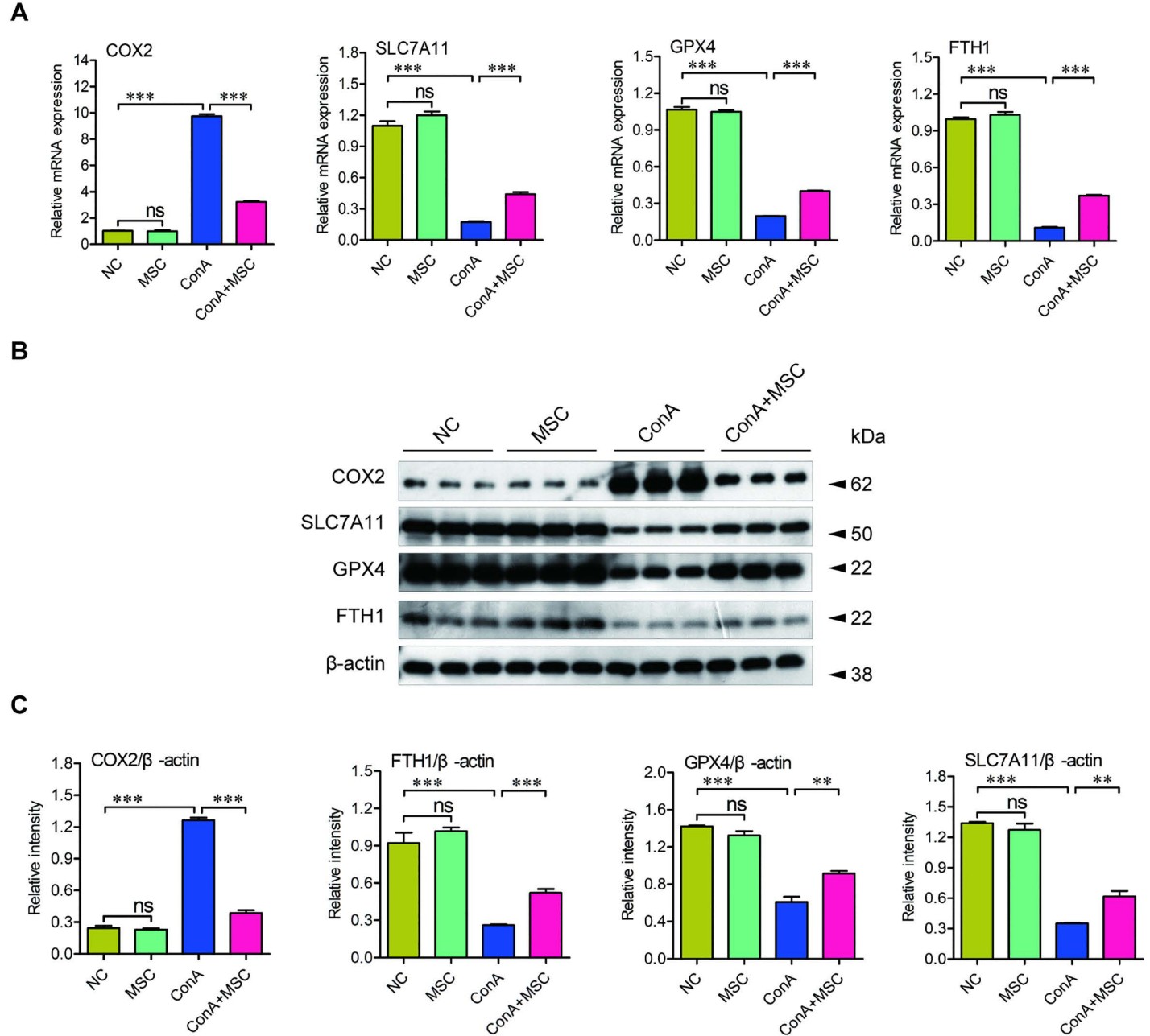

**Fig 7. hUC-MSCs regulate expression of ferroptosis-related genes and proteins.** (A) qPCR analysis of COX2, SLC7A11, GPX4, and FTH1 expression in liver tissues among the groups. n = 6. (B) Representative Western blots of COX2, SLC7A11, GPX4, and FTH1. (C) Quantification of band intensities normalized to β-actin. n = 3; **p < 0.01, ***p < 0.001, ns, not significant. Two-way ANOVA followed by Tukey's HSD was utilized to assess the statistical significance.

## 4. Discussion

In this study, we demonstrate the potent therapeutic effects of hUC-MSCs in a murine model mimicking AIH induced by ConA. Our findings reveal that hUC-MSC administration not only significantly improves survival outcomes and liver

function but also mitigates hepatic tissue damage and apoptosis. Notably, mechanistic insights from metabolomic profiling and molecular analyses suggest that suppression of ferroptosis is a key component of hUC-MSC-mediated hepatoprotection.

MSCs are increasingly recognized for their dual regenerative and immunomodulatory functions. In the context of liver injury, these cells may exert protective effects through multiple pathways, including repressing the secretion of inflammatory cytokines, inhibiting immune cell activation, and promoting tissue repair [18,19]. However, the mechanisms by which MSCs confer protection in autoimmune-mediated liver disease remain insufficiently understood.

Our untargeted metabolomic analysis provided a systems-level view of the metabolic changes associated with AIH and MSC treatment. The identification of ferroptosis-related metabolic pathways, including altered glutamate metabolism, oxidative stress markers, and ABC transporter function [20–23], pointed to ferroptosis as a likely target of MSC action. This was supported by overlapping DEMs associated with Glu derivatives and peptide metabolism, both linked to iron homeostasis and redox balance [24,25].

Ferroptosis, characterized by lipid peroxidation and iron-dependent oxidative damage, has emerged as a critical mechanism of cell death in liver disease [26,27]. In our study, ConA exposure induced classic features of ferroptosis, including increased hepatic MDA, GSSG, Glu, and $Fe^{2+}$ levels, along with a decrease in GSH. These changes were significantly attenuated by hUC-MSC treatment. Moreover, the expression of key ferroptosis-regulatory proteins was partially restored toward homeostatic levels, with suppression of the pro-ferroptotic marker COX2 and upregulation of protective proteins such as SLC7A11, GPX4, and FTH1. These results suggest that hUC-MSCs mitigate liver injury at least in part by modulating the ferroptosis pathway.

While these findings offer compelling mechanistic insights, further investigation is necessary to determine whether hUC-MSCs inhibit ferroptosis primarily through paracrine signaling pathways or via the direct transfer of antioxidant molecules. The precise identity of the bioactive factors responsible for modulating ferroptosis remains unknown and warrants deeper exploration. Moreover, it would be of considerable interest to assess whether extracellular vesicles (EVs) derived from hUC-MSCs can reproduce the ferroptosis-suppressive and hepatoprotective effects of their parent cells in AIH-induced liver injury, as has been observed in other models of liver damage [28–30]. If EVs exhibit comparable or enhanced efficacy, their use may present a more targeted and scalable therapeutic strategy, given their stability, low immunogenicity, and ease of storage and handling [31,32].

It is noteworthy that our study has some limitations. For instance, while ferroptosis suppression was clearly associated with improved outcomes, causal relationships require confirmation through targeted inhibition or enhancement of ferroptosis in vivo.

In summary, our study highlights the therapeutic promise of hUC-MSCs in treating AIH, demonstrating their ability to protect hepatocytes, at least in part, through the inhibition of ferroptosis. These findings enhance our understanding of the mechanisms underlying stem cell interventions in autoimmune liver diseases such as AIH.

**hUC-MSCs:** Multipotent stem cells isolated from Wharton's jelly of the umbilical cord, with immunomodulatory and regenerative properties.

**Ferroptosis:** A regulated form of cell death driven by iron-dependent lipid peroxidation, involving glutathione depletion and oxidative stress.

**ConA:** A plant lectin used experimentally to induce T-cell-mediated liver injury, mimicking autoimmune hepatitis in murine models.

**OPLS-DA:** A multivariate statistical method used to classify groups and identify biomarkers by separating predictive and non-predictive variation in metabolomic data.

**ABC Transporters:** Membrane proteins involved in metabolite transport; linked to drug resistance and ferroptosis regulation.

**EVs:** Membrane-bound particles secreted by cells (e;g;, extracellular vesicles), implicated in intercellular communication and potential MSC-derived therapeutics.

## Supporting information

**S1 Fig. Characterization of hUC-MSCs.** (A) Cell viability of freshly isolated hUC-MSCs was assessed by trypan blue exclusion using an automated cell counter. (B) Flow cytometry analysis of hUC-MSC surface markers. Cells exhibited high expression of MSC markers CD73, CD90, and CD105, while hematopoietic lineage markers (CD11b, CD19, CD34, CD45, HLA-DR) were nearly absent. Isotype controls were used to define positivity thresholds.
(TIF)

**S2 Fig. Hepatic homing of hUC-MSCs in control and ConA-injured mice.** (A) Representative bright-field and fluorescence images showing successful transduction of hUC-MSCs with lentivirus encoding ZsGreen and Luciferase. (B) Flow cytometric analysis of infected and control hUC-MSCs. (C, D) qPCR analysis of ZsGreen and Luciferase mRNA expression in liver tissues from mice with the indicated treatment. $n = 6$; ***$p < 0.001$. Two-way ANOVA followed by Tukey's HSD was utilized to assess the statistical significance.
(TIF)

**S1 File. Uncropped Western blot for Figure 7B.**
(DOCX)

**S2 File. Uncropped Western blot for Figure S2C.**
(DOCX)

**S3 File. Raw data for the study.**
(ZIP)

## Author contributions

**Conceptualization:** Yaqin Li, Yihua Chen, Youhua Xu, Guangdong Tong, Wei V. Zheng.

**Data curation:** Yaqin Li, Bing Liu, Ling Guo, Luoshi Zhang.

**Formal analysis:** Yaqin Li, Bing Liu, Tao Zhou, Weizhao Tong.

**Funding acquisition:** Yihua Chen, Youhua Xu, Guangdong Tong, Wei V. Zheng.

**Investigation:** Yaqin Li, Guoxin Hu, Xuanqiu He.

**Methodology:** Yaqin Li, Guoxin Hu, Xuanqiu He.

**Project administration:** Yihua Chen, Youhua Xu, Guangdong Tong, Wei V. Zheng.

**Resources:** Guoxin Hu, Ling Guo, Luoshi Zhang.

**Software:** Bing Liu, Tao Zhou.

**Supervision:** Yihua Chen, Youhua Xu, Guangdong Tong, Wei V. Zheng.

**Validation:** Bing Liu, Weizhao Tong.

**Visualization:** Tao Zhou.

**Writing – original draft:** Yaqin Li.

**Writing – review & editing:** Yihua Chen, Youhua Xu, Guangdong Tong, Wei V. Zheng.

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
