## [Decision Letter · Decision Letter 0]

23 Jul 2025

Dear Dr. Zheng,

Thank you for submitting your manuscript to PLOS ONE. After careful consideration, we feel that it has merit but does not fully meet PLOS ONE’s publication criteria as it currently stands. Therefore, we invite you to submit a revised version of the manuscript that addresses the points raised during the review process.

We look forward to receiving your revised manuscript.

Kind regards,

Junzheng Yang

Academic Editor

PLOS ONE

Journal Requirements:

“1. National Natural Science Foundation of China, 82371568

2. Shenzhen Science and Technology Program, JCYJ20220818102818039.

3. Shenzhen Biomedical Industry Major Public Service Platform and Core Technology Research Special Support Plan�XMHT20220104048.

4. BYD Charity Foundation: PUSH-BYD JCYJ20220401�PUSH-BYD JCYJ20220.

5. Medical Scientific Research Foundation of Guangdong Province, A2024220

6.Peking University Shenzhen Hospital Research Foundation, JCYJ 2021014”

4. We note that your Data Availability Statement is currently as follows: All relevant data are within the manuscript and its Supporting Information files

Reviewers' comments:

Reviewer's Responses to Questions

**Comments to the Author**

1. Is the manuscript technically sound, and do the data support the conclusions?

Reviewer #1: Yes

Reviewer #2: Partly

2. Has the statistical analysis been performed appropriately and rigorously?

Reviewer #1: No

Reviewer #2: N/A

3. Have the authors made all data underlying the findings in their manuscript fully available?

Reviewer #1: Yes

Reviewer #2: Yes

4. Is the manuscript presented in an intelligible fashion and written in standard English?

Reviewer #1: Yes

Reviewer #2: Yes

Reviewer #1: Page14/line5: `randomly assigned into four groups (n = 3 per group)`, the number in each group is too small and impossible to run the statistics. Please add more animals in each group and reevaluate the experiment.

Page14/line8-9: how do you determine the cell number to inject. 1x10^6 cells per animal is too big amount. Is there any scientific reasons or references to support choosing this number. Please provide BW and gender in each group.

After cell injection, how many hUC-MSCs were reached and retained in the liver.

Stastics: "GraphPad Prism8" is application name, please provide what kind of statsitic was used in each experiment.

Figue2: since upper and lower panel did not match, it is confusing to understand. In panel B, what yellow dots and white dots are indicating.

Reviewer #2: the manuscript presents a comprehensive and well-structured investigation into the therapeutic potential of human umbilical cord-derived mesenchymal stem cells (hUC-MSCs) in autoimmune hepatitis (AIH), particularly focusing on the suppression of ferroptosis as a key protective mechanism

1. Abstract Revision – Improve Specificity and Flow

Current Issue: The abstract is informative but lacks precision in quantitative terms and slightly overstates mechanistic conclusions.

2. Sample Size Limitation, Only n = 3 per group in key experiments.

While preliminary data is still valuable, such a small number limits statistical power and generalizability. A sample size of 3 severely underpowers the study and affects statistical reliability.

3. Mechanistic Causality Needs Clarification

While ferroptosis marker modulation is demonstrated, causal inference remains correlative. so the paper lack of Causal Validation of Ferroptosis Pathway

4. Missing Controls: A ConA + vehicle control group (without MSCs) for comparison in qPCR and Western blot analysis should be explicitly stated and shown.

5. There is minimal detail on how hUC-MSCs were characterized for identity, purity, and viability before administration.

6. Clarify Timing of Sampling Post-ConA. Inconsistent Statements: “24h after ConA injection” is mentioned in methods, but survival data are observed up to 18h.

**Do you want your identity to be public for this peer review?** For information about this choice, including consent withdrawal, please see our Privacy Policy

Reviewer #1: No

Reviewer #2: **Yes: ** Ramada R. K.

---

## [Author Response · Author response to Decision Letter 1]

4 Sep 2025

Reviewer #1: Page14/line5: `randomly assigned into four groups (n = 3 per group), the number in each group is too small and impossible to run the statistics. Please add more animals in each group and reevaluate the experiment.

Response: We appreciate the reviewer’s concern. We repeated the experiments in an additional six mice and incorporated the new data into the revised figures. One important clarification regarding the repeated experiments is that the two GLU measurements were performed using different assay kits. The original data were obtained with the MEIMIAN Glutamic Acid (Glu) Assay Kit (Cat. No. ADS-F-AJS007), whereas the repeat experiments used the Elabscience Glu Colorimetric Assay Kit (Cat. No. E-BC-K903-M). Owing to differences in the calculation methodologies between the two kits, the absolute values varied. Importantly, however, the intergroup trends remained consistent across both experiments.

Page14/line8-9: how do you determine the cell number to inject. 1x10^6 cells per animal is too big amount. Is there any scientific reasons or references to support choosing this number. Please provide BW and gender in each group.

After cell injection, how many hUC-MSCs were reached and retained in the liver.

Response: The injection of 106 MSCs is a commonly used dosage in similar studies. Numerous publications have employed this dosage (e.g., PMID: 22752635, PMID: 33757765, PMID: 31133062). In the revised manuscript, we have included the baseline body weight of the mice (approximately 22-25 grams per mouse) before the experiments. The gender of the animals (female) was clearly stated in the original manuscript. Regarding the amount of hUC-MSCs homing to the liver, we did not specifically quantify the exact cell number. However, we generated GFP-labeled hUC-MSCs and injected them into mice under different treatment conditions. As shown in the revised Figure S2, hUC-MSCs were able to reach and be retained in the liver, with higher levels observed in injured liver tissues. We believe these results provide sufficient evidence to support our conclusion.

Stastics: "GraphPad Prism8" is application name, please provide what kind of statsitic was used in each experiment.

Response: The methods for statistical analyses had been clearly described in the figure legends.

Figue2: since upper and lower panel did not match, it is confusing to understand. In panel B, what yellow dots and white dots are indicating.

Response: Thank you for your comment. In the revised Figure 2, we have added dashed boxes in parts A and B to clearly indicate the correspondence between the upper and lower panels. The lower panels present magnified views of the specific regions highlighted in the upper panels. Yellow dots mark TUNEL-positive cells. The white areas are likely due to exceptionally strong apoptotic fluorescence signals from individual cells.

Reviewer #2: the manuscript presents a comprehensive and well-structured investigation into the therapeutic potential of human umbilical cord-derived mesenchymal stem cells (hUC-MSCs) in autoimmune hepatitis (AIH), particularly focusing on the suppression of ferroptosis as a key protective mechanism

1. Abstract Revision – Improve Specificity and Flow

Current Issue: The abstract is informative but lacks precision in quantitative terms and slightly overstates mechanistic conclusions.

Response: Thank you for your comment. We have revised the Abstract to enhance its conciseness and precision, ensuring that it now more accurately reflects the key findings and perspectives of our study. We appreciate your feedback, which has been invaluable in improving the clarity of our manuscript.

2. Sample Size Limitation, Only n = 3 per group in key experiments.

While preliminary data is still valuable, such a small number limits statistical power and generalizability. A sample size of 3 severely underpowers the study and affects statistical reliability.

Response: Thank you for your comment. We have repeated the experiments using six additional mice and incorporated the new results into the revised figures.

3. Mechanistic Causality Needs Clarification

While ferroptosis marker modulation is demonstrated, causal inference remains correlative. so the paper lack of Causal Validation of Ferroptosis Pathway

Response: We understand the reviewer’s concern and have acknowledged this limitation in the Discussion section. Indeed, the exploration of ferroptosis-related mechanisms in this study remains relatively preliminary. While our current work primarily demonstrates an association between MSC-based therapy and ferroptosis inhibition, a more in-depth investigation into the underlying mechanisms will be pursued in a separate project using a broader range of experimental approaches. To further address this issue in the present study, we plan to conduct additional rescue experiments, such as the application of ferroptosis agonists, to clarify the causal relationship between the protective effects of MSCs and ferroptosis inhibition. We will publish these data in the future.

4. Missing Controls: A ConA + vehicle control group (without MSCs) for comparison in qPCR and Western blot analysis should be explicitly stated and shown.

Response: We suspended the MSCs in PBS and administered the same volume of PBS, used for MSC suspension, to the mice in the ConA group, serving as the appropriate control as suggested by the reviewer. This information has been included in the revised manuscript.

5. There is minimal detail on how hUC-MSCs were characterized for identity, purity, and viability before administration.

Response: We apologize for the missing details and have included them in the revised manuscript and Figure S1.

6. Clarify Timing of Sampling Post-ConA. Inconsistent Statements: “24h after ConA injection” is mentioned in methods, but survival data are observed up to 18h.

Response: The survival data (Figure A) show that all mice in the ConA group died within 18 hours, whereas all mice in the other groups survived for at least 24 hours, as indicated by the red line at the top.

---

## [Decision Letter · Decision Letter 1]

10 Oct 2025

Dear Dr. Zheng,

Thank you for submitting your manuscript to PLOS ONE. After careful consideration, we feel that it has merit but does not fully meet PLOS ONE’s publication criteria as it currently stands. Therefore, we invite you to submit a revised version of the manuscript that addresses the points raised during the review process.

We look forward to receiving your revised manuscript.

Kind regards,

Ramada Rateb Khasawneh

Academic Editor

PLOS ONE

Journal Requirements:

3. Please provide the original blot images for Figure 2S.

4. Please clarify the reasons for the authorship changes and specify why the changes were not made at initial submission.

Additional Editor Comments (if provided):

It is a good paper overall, but there are a few technical points I have raised.

The abstract mentions “metabolomic analysis revealed modulation of pathways related to ferroptosis,” but does not specify whether pathway enrichment or specific metabolites were validated. A clearer statement of the level of evidence (correlation vs. validation) would strengthen accuracy.

The abstract implies that hUC-MSCs inhibit ferroptosis as the mechanistic basis of protection (“through inhibiting the ferroptosis pathway”).

Unless ferroptosis inhibition was directly tested (e.g., by using ferroptosis activators/inhibitors or rescue assays), it would be more accurate to phrase this as: “...potentially through modulation of ferroptosis-related pathways.”

“hUC-MSCs” defined twice — keep only once

The ConA model indeed induces T-cell–mediated immune hepatitis, but calling it “autoimmune hepatitis” should be qualified — it is an AIH-like murine model, not true autoimmune hepatitis .. please correct this

Reviewers' comments:

Reviewer's Responses to Questions

**Comments to the Author**

Reviewer #1: All comments have been addressed

Reviewer #2: All comments have been addressed

Reviewer #3: All comments have been addressed

2. Is the manuscript technically sound, and do the data support the conclusions?

Reviewer #1: Yes

Reviewer #2: Yes

Reviewer #3: Yes

3. Has the statistical analysis been performed appropriately and rigorously?

Reviewer #1: Yes

Reviewer #2: I Don't Know

Reviewer #3: Yes

4. Have the authors made all data underlying the findings in their manuscript fully available?

Reviewer #1: Yes

Reviewer #2: Yes

Reviewer #3: Yes

5. Is the manuscript presented in an intelligible fashion and written in standard English?

Reviewer #1: Yes

Reviewer #2: Yes

Reviewer #3: Yes

Reviewer #1: Authors addressed reviwer1's concern adequately. Reviewer1 has no further question on this manuscript.

Reviewer #2: The author has carefully and comprehensively addressed all the reviewers’ comments and concerns. All issues raised during the review process have been adequately resolved, and the revised manuscript now meets the journal’s standards in terms of scientific quality, clarity, and presentation. Therefore, I believe the paper is now suitable for publication in its current form.

Reviewer #3: This manuscript is thorough and technically sound, and it is clear that the authors have addressed all previous reviewers' concerns.

**Do you want your identity to be public for this peer review?** For information about this choice, including consent withdrawal, please see our Privacy Policy

Reviewer #1: No

Reviewer #2: No

Reviewer #3: No

---

## [Author Response · Author response to Decision Letter 2]

11 Oct 2025

3. Please provide the original blot images for Figure 2S.

Response: Thanks for the comment. We have provided the original blot image for panel C of Figure S2 in the separate file "uncropped western blot-2". Please note that this is the only panel from this figure S2 for which a new original blot is required.

4. Please clarify the reasons for the authorship changes and specify why the changes were not made at initial submission.

Response: Thanks for the comment.During the revision process, we have added Mr. Luoshi Zhang as a co-author to the manuscript. His substantial contributions to performing the critical experiments outlined in the response letter and his analysis of the new data fully meet the criteria for authorship.

Additional Editor Comments (if provided):

It is a good paper overall, but there are a few technical points I have raised.

The abstract mentions “metabolomic analysis revealed modulation of pathways related to ferroptosis,” but does not specify whether pathway enrichment or specific metabolites were validated. A clearer statement of the level of evidence (correlation vs. validation) would strengthen accuracy.

Response: Thanks for the advice. We have rephrased the sentence as follows: “Metabolomic and ontology analyses of mouse liver tissue samples revealed that hUC-MSCs treatment altered the levels of metabolites (Glu derivatives and peptides) functionally associated with ferroptosis-related pathways.” Since we didn’t perform validation, emphasizing ontology analysis should be sufficient to clarify this.

The abstract implies that hUC-MSCs inhibit ferroptosis as the mechanistic basis of protection (“through inhibiting the ferroptosis pathway”).

Unless ferroptosis inhibition was directly tested (e.g., by using ferroptosis activators/inhibitors or rescue assays), it would be more accurate to phrase this as: “...potentially through modulation of ferroptosis-related pathways.”

“hUC-MSCs” defined twice — keep only once

Response: We appreciate the advice. The relevant text has been modified accordingly. Yet, we can’t find a duplicated definition of hUC-MSCs in the Abstract. Following your suggestion, we have removed the redundant definition of hUC-MSCs from the Introduction, while retaining the single definition in the Abstract.

The ConA model indeed induces T-cell–mediated immune hepatitis, but calling it “autoimmune hepatitis” should be qualified — it is an AIH-like murine model, not true autoimmune hepatitis .. please correct this

Response: Thanks for the advice. We have made the modification in the revised manuscript. As suggested, we have changed "ConA-induced AIH mouse model" to "ConA-induced AIH-like mouse model" throughout the manuscript.

---

## [Decision Letter · Decision Letter 2]

4 Nov 2025

Human Umbilical Cord-Derived Mesenchymal Stem Cells Alleviate Autoimmune Hepatitis by Inhibiting Hepatic Ferroptosis

PONE-D-25-34339R2

Dear Dr. Zheng,

We’re pleased to inform you that your manuscript has been judged scientifically suitable for publication and will be formally accepted for publication once it meets all outstanding technical requirements.

Kind regards,

Ramada Rateb Khasawneh

Academic Editor

PLOS ONE

Additional Editor Comments (optional):

nice paper .. Good Luck

Reviewers' comments:

Reviewer's Responses to Questions

**Comments to the Author**

Reviewer #1: All comments have been addressed

Reviewer #2: All comments have been addressed

2. Is the manuscript technically sound, and do the data support the conclusions?

Reviewer #1: Yes

Reviewer #2: Yes

3. Has the statistical analysis been performed appropriately and rigorously?

Reviewer #1: Yes

Reviewer #2: I Don't Know

4. Have the authors made all data underlying the findings in their manuscript fully available?

Reviewer #1: Yes

Reviewer #2: Yes

5. Is the manuscript presented in an intelligible fashion and written in standard English?

Reviewer #1: Yes

Reviewer #2: Yes

Reviewer #1: Authors addressed reviewer’s concerns.Reviewer has no further comment. I agree to publish this manuscirpt.

Reviewer #2: The article appears to be in excellent condition following the recent revisions. The authors have effectively addressed the previous comments and substantially improved the clarity, organization, and overall scientific quality of the manuscript. In its current form, the paper seems well-prepared and suitable for publication.

**Do you want your identity to be public for this peer review?** For information about this choice, including consent withdrawal, please see our Privacy Policy

Reviewer #1: No

Reviewer #2: No

---

## [Editor Report · Acceptance letter]

PONE-D-25-34339R2

PLOS ONE

Dear Dr. Zheng,

I'm pleased to inform you that your manuscript has been deemed suitable for publication in PLOS ONE. Congratulations! Your manuscript is now being handed over to our production team.

Kind regards,

on behalf of

Dr. Ramada Rateb Khasawneh

Academic Editor

PLOS ONE